# A New Methodology to Design Sustainable Archimedean Screw Turbines as Green Energy Generators

**DOI:** 10.3390/ijerph17249236

**Published:** 2020-12-10

**Authors:** Mar Alonso-Martinez, José Luis Suárez Sierra, Juan José del Coz Díaz, Juan Enrique Martinez-Martinez

**Affiliations:** Construction and Manufacturing Engineering Department, University of Oviedo, EDO-7 Campus de Viesques, 33204 Gijón (Asturias), Spain; suarezsjose@uniovi.es (J.L.S.S.); juanjo@constru.uniovi.es (J.J.d.C.D.); quique@constru.uniovi.es (J.E.M.-M.)

**Keywords:** Archimedean screw turbine, green energy, sustainable construction, resilient structures

## Abstract

Current energy demand and climate target plans are leading to green energy facilities which are efficient and sustainable. Archimedean screw turbines (ASTs) are used to generate hydroelectricity in low heads. They have been manufactured and installed worldwide. However, there is a lack of knowledge about how to design them efficiently. In this study, the performance of ASTs is analyzed using an analogy between ASTs and bucket elevators. Based on this analogy, a theoretical hypothesis on how to produce efficient ASTs is proposed. The new methodology for the design of ASTs is based on two considerations: the filling level of the AST buckets must be 85% and the increase of leakage losses must be minimized. This hypothesis is numerically and experimentally studied. Two experimental prototypes were developed and installed in the north of Spain. The numerical and experimental results are provided. A discussion comparing the results of this work and other results from the literature is presented. Finally, conclusions are drawn from this work that contribute to the improvement of AST technology as a sustainable facility to generate green energy.

## 1. Introduction

High energy consumption is one of the worries of our society. Energy demand increases every year and generation of green energy is essential to country growth that minimizes environmental impacts. Although there are many types of systems that can be used to generate green energy, sustainable infrastructures are needed to achieve sustainable development. Archimedean screw turbines (ASTs) are low-head hydropower generators suitable for industrial applications. 

Low-head hydropower develops hydroelectric power using low heads, usually less than 20 m high. It is widely known as a renewable energy source. This highly efficient hydropower is inexpensive to manufacture and maintain. It reduces greenhouse gas emissions with minimal environmental impact. Microhydropower plants have been designed and studied in depth because of their advantages over traditional fossil fuels [1].

The first device used to generate electricity from running water was the waterwheel in the Fox River, Wisconsin, in 1882 [1]. Currently, hydropower devices which generate less than 100 kW are known as microhydropower generators. The common range of energy is from 5 kW to 100 kW but there are a great number of locations where small devices generating less than 5 kW could be installed [2]. 

There are many types of low-head devices for low-head hydropower applications, such as traditional waterwheels, Kaplan, Francis and Pelton turbines and ASTs. The AST was patented in 1991 by Karl-August Radlik [3]. It has gained popularity recently due to important advantages over alternatives, such as its suitability for low heads, the by “the simple construction process”needed for its installation and its reduced environmental impact [4,5,6,7]. Although the Archimedes screw was developed for pumping water from lower to higher levels, the use of this device in recent years focuses on the generation of sustainable energy. The AST is efficient in locations with low head and low flow [5]. The most relevant advantages of the AST are its low cost of manufacturing, installation and maintenance and its low environmental impact. In addition, the construction process needed is very simple and fish friendly [4,5,6,7]. Due to the low rotational speed and the atmospheric pressure, most fish species are able to pass through the turbine with no injuries. Although the benefits of ASTs make them a sustainable facility to generate green energy, adequate design methodologies must be followed. 

Most previous studies model and test ASTs in small-scale prototypes in the laboratory. In 2012 Lashofer et al. published an overview of the AST technology including design and operation. An in-depth study of 74 ASTs installed in 71 locations across Europe was published in the same year [3]. The main parameters of AST geometry, the speed rotation, the civil construction time and the most relevant operating issues were presented. An important contribution of this report [3] was the provided AST design values established by manufacturers and operators from functioning AST plants in Europe. The plant efficiency was obtained as a function of load ratio, which is the actual flow at the time of measurement over the design flow. The results of the report show very similar efficiency patterns for all the plants studied, with peak efficiencies around 75%. Other studies find similar efficiencies of between 60% and 80% [2,4,8]. Most of the references agree that leakage losses are one of the most influential parameters in the efficiency of turbines [2]. Leakage losses are mainly due to the gap between the flights of the screw and the trough, which allows the screw to rotate.

Although there are many AST facilities around the world which have been tested for efficiency, there is a lack of understanding about how ASTs work and how to design them more efficiently. There are some authors that explain the AST as a mechanism similar to a waterwheel [8]. This similarity has not been analytically proved. Most of the studies are based on experimental experiences [3,5,6,7,9,10]. However, to understand the technology and optimize its efficiency, theoretical hypotheses concerning AST performance are needed. The losses in this kind of turbine and their influence on efficiency must be studied in depth. 

In this study, the performance of ASTs is analyzed using an analogy between ASTs and bucket elevators. Based on this analogy, AST performance is explained, including the fundamentals required to obtain high efficiency, approximately 80%, in real facilities. Previously published efficiency curves of ASTs are discussed. A new methodology to design ASTs based on the leakage losses curve is presented. The authors establish that leakage losses follow a linear trend, which is a function of the ratio of flow (actual flow over maximum flow). This methodology was tested in two prototypes developed and installed in the north of Spain. Key parameters to design efficient ASTs are revealed and the sustainability of this technology is improved. A discussion comparing the efficiencies obtained in this work and the most relevant contributions from the literature is presented. Finally, conclusions are drawn that contribute to the development of sustainable facilities to generate green energy.

## 2. Materials and Methods 

### 2.1. Description of the Archimedes Screw Turbine Technology 

In this work, an analogy between ASTs and bucket elevators is used to identify the most influential parameter on total efficiency. The total efficiency of the AST is divided into hydraulic, mechanical and electrical efficiencies. In this study, the mechanical and electrical efficiencies were estimated as 90%. The present analysis only aims at improving the hydraulic efficiency of ASTs, which can improve the total efficiency by about 10%. Figure 1 shows the mechanisms of ASTs and bucket elevators, including their main design parameters, which are as follows:Z is the height of the head;Q, in m^3^/s, is the total flow passing through the hydraulic machine, the AST or the bucket elevator;Q_leak_, in m^3^/s, is the leakage loss flow due to leakage between two consecutive levels;Q_overflow_, in m^3^/s, is the overflow loss when the design flow is exceeded. These losses appear in ASTs although they are not included in Figure 1a.

The design parameters of ASTs, shown in Figure 1a, are the following:
P, in m, is the pitch of the screw;L, in m, is the length of the screw.

The design parameter of the bucket elevator, shown in Figure 1b, is R, the radius of the pulley used to move the belt, measured in m.

For an ideal machine where the maximum flow, Q, is the design flow, the efficiency will be 1. However, in real machines variable flows and losses flows must be taken into account to determine the real efficiency. Three operation conditions were used to study AST.

(a)AST operates at the design flow, Qmax.:

When a bucket elevator is operating under design flow conditions, the buckets are full and the efficiency is mainly determined by the losses. Leakage losses in the buckets of the elevator are due to imperfections, holes or defects that are avoided during its manufacturing process and controlled during its operation. However, leakage losses in ASTs are mainly due to the gap between the flights of the screw and the trough. These losses are unavoidable in order to ensure the good performance of the AST. For this reason, leakage flow was analyzed in this study and its influence on the efficiency of the mechanism was determined. The unavoidable leakages were determined in this work using a small-scale prototype in experimental tests for different flow levels.

(b)AST operates below the design flow:

If the total flow is less than the design flow and the rotation velocity is the same as the velocity of the design flow, the efficiency is lower. This is easy to understand in the bucket elevator because below the design flow the capacity of the buckets is lower and the torque of the mechanism is also lower. This reduction of torque leads to a lower efficiency and a lower generation of energy. This effect also takes place in ASTs. In this study, the influence of the volume of water on the torque, and consequently on the efficiency, is studied.

(c)AST operates above the design flow, overflow:

The leakage due to overflow refers to the water losses due to overfilling when the total flow is exceeded. In a bucket elevator overflow occurs due to overfilling of the buckets. In ASTs these leakages also appear although their effect on the efficiency is lower than the effect due to leakage between the flights and the trough. 

The hypothesis of this work defines leakage loss as a linear trend as follows. Unavoidable leakages were experimentally obtained with the AST stopped. Then, the increase of leakage losses was determined as a function of the rotation speed. This is presented in Equation (1).
(1)QleakQmax=Ls+LsΩ60QQmax
where:
*Q_leak_* is the loss due to leakage (flow in m^3^/s);*Q_max_* is the maximum flow (m^3^/s);*L_s_* is the static leakage loss (%);Ω is the rotation speed of the screw (rpm);*Q* is the actual flow through the AST (m^3^/s).

This hypothesis was checked against efficiency curves published by other authors and manufacturers [11,12,13]. In 2015, Charisiadis published a review of the Archimedean screw as a low-head generator considered as an eco-friendly technology. ASTs and their benefits were also described in a booklet that collected data from previous works. In this publication, an AST efficiency curve was included and compared with traditional generators such as waterwheels, propellers and Kaplan and Francis turbines [11]. Another study compared different types of hydropower turbines [12]. The authors explained why AST efficiency under real conditions is lower than that claimed by the manufacturers. They also included efficiency curves of ASTs. The last efficiency curve included in this work is from the manufacturer Roncuzzi Renewable Energy [13]. In its publication, details about the civil work, the benefits of hydropower screws and performance were included. 

The AST is a volumetric turbine and thus based on volumetric efficiency (see Equation (2)). The authors obtained the leakage losses curve from the efficiency curve of the AST.
(2)ηv=1−QleakQmaxQQmax

Using this hypothesis, and considering the information previously published, the authors determined the leakage losses curve of the AST from the references [11,12,13]. 

The real leakage losses can be obtained in ASTs which have already been installed and thus the adjustment of the linear trendline proposed in this work can be determined. 

This hypothesis provides a new methodology to design efficient ASTs. The key points of the hypothesis used to design high-efficiency ASTs are the following:The AST must be understood as a bucket elevator and designed to take into account the filling level.The AST must be designed to ensure as low a percentage of leakage losses as possible.

In order to prove this hypothesis, numerical and experimental analyses were developed.

Firstly, the filling level of the buckets was studied through numerical models using the finite element method (FEM). In addition, two experimental prototypes were designed following this methodology based on linear leakage losses. Experimental data from the prototypes reveal the efficiency of this kind of AST and its leakage losses. 

### 2.2. Study of Theoretical Performance of the AST through Numerical Models using the Finite Element Method

The complex geometry of the flights of a screw makes its analytical study very difficult. Numerical models using FEM are a powerful tool to study those complex geometries and the filling level and its effect on the efficiency of the turbine.

The numerical models presented in this work analyze the effect of the filling of the buckets on the torque. The torque is directly related to the efficiency of the AST. The higher the torque is, the more power is generated. The analysis was done considering the design flow.

The geometrical model of the AST is shown in Figure 2a. To study the effect of the filling of the buckets, only one bucket was studied. The numerical model is shown in Figure 2b. The type of finite structural element used to simulate the flights was SHELL93 [14]. This element is suitable for modeling curved shells. It has six degrees of freedom at each node: translations in x, y and z and rotations about x, y and z.

The numerically analyzed characteristics of the AST are detailed in Table 1. 

Four numerical models were developed to study the following levels of filling: 41%, 52%, 69% and 85%. The boundary conditions of the numerical model are shown in Figure 3. 

Node A is in the axis of the screw. Displacements and rotations at node A were avoided. The filling of the bucket was modeled by applying the fluid pressure normal to the surface. The surface was obtained from the geometrical model and the value of the normal pressure depends on the filling level.

Although the theoretical performance of an ideal AST would provide an efficiency of 100%, a real mechanism has losses due to leakage, overflow and other factors. To study the real efficiency of the AST two experimental prototypes were manufactured and installed in the north of Spain. 

### 2.3. Real Performance of the AST Using Experimental Prototypes

Two prototypes were developed and installed in two different locations. A small-scale prototype was installed in the Atlantic Botanic Garden of Gijón (Spain) in 2018 [15] (see Figure 4a) and a real-scale AST was installed in Barreda hydroelectric plant in Torrelavega (Spain) [16] (see Figure 4b).

The experimental setup recalculates the flow to measure the flow, torque and rotational speed of the AST prototypes. The system is shown in Figure 5.

#### 2.3.1. Small-Scale Prototype

The AST installed in Gijon was an experimental prototype designed for research purposes in the University of Oviedo. It was situated in a small lake and used to generate electricity to supply energy for an ultrasound system to avoid eutrophication issues. The head was very small, only 1.2 m high, and the design flow was 1.20 m^3^/s. However, the AST prototype generated 1.2 kW and helped to supply the ultrasound control with energy efficiently and in a fish friendly and sustainable way.

The screw and the trough were made of steel and accurately manufactured. The gap between the flights and the trough was 3 mm. The maximum rotation speed was 73 rpm. The hypothesis presented in this work, Equation (1), was used to minimize leakage losses and obtain an efficient AST.

This small-scale prototype was tested under different flows to analyze the efficiency of the AST under real conditions.

The main design parameters of the small-scale AST are shown in Table 2. 

#### 2.3.2. Real-Scale Prototype

A real-scale prototype was also designed based on the leakage losses and their influence on the efficiency. The prototype was installed in a low head, less than 2 m high, of the Saja River. Two ASTs, able to generate 35 kW/turbine, were installed in the facility to generate 70 kW. The design flow was 2.50 m^3^/s per turbine. The maximum rotation speed was 50 rpm although there was a variable speed control. 

The screw was made of steel and the trough was a simple civil work made of concrete.

The main parameters of the real-scale AST are shown in Table 3. 

## 3. Results

### 3.1. Numerical Results

Force reactions obtained in node A provided the torque of the screw for the four levels of filling studied. Figure 6 shows these results including the linear trendline. The adjustment of the trendline based on R-squared values was very good and showed the expected linear trend. For the design flow, the maximum theoretical torque was obtained when buckets were full. This effect, which is clear in a bucket elevator, is not so obvious in the AST design. In a bucket elevator, the torque is the weight of the bucket times the radious of the belt pulley. However, the concept of buckets and of filling in ASTs has not been clearly explained in previous research. 

The results of this numerical analysis prove that the torque of the AST was directly proportional to the filling volume of the buckets. Filling volume and flow were also directly related by the rotation speed of the screw. Therefore, to obtain maximum efficiency the rotation speed of the screw plays a very important role.

### 3.2. Experimental Results

#### 3.2.1. Small-Scale Prototype Results

Tests done on the small-scale prototype demonstrated the efficiency of an AST for a very low head with different flows. The system was designed to work with the buckets 85% full. This value was identified as the optimum filling level in the experimental tests. Above that value, overflow occurs, as was seen in the experimental tests and in the decrease in efficiency.

Figure 7 shows the efficiency curve of this specific prototype based on the relation between the real flow measured and the maximum flow. 

Leakage losses were also obtained from the small-scale prototype. Figure 8 shows the percentage of leakage losses in the small-scale prototype. Static leakage losses were measured with the screw stopped. It was demonstrated that 5% losses were unavoidable. The leakage losses increased slowly, confirming the hypothesis established about the design of the AST. 

#### 3.2.2. Real-Scale Prototype Results

Results obtained for the real-scale prototype from Barreda hydroelectric power plant demonstrated high efficiencies, usually above 80%. This experimental prototype was designed to work with a bucket 85% full. In this prototype, the filling level was ensured using an active speed control. This active speed control checks the torque given by the AST every 10 min. The control varies the speed using a converter and an algorithm is provides five torque/speed combinations. All the combinations are immediately compared and the control chooses the highest power available and sets the speed to obtain that power as the nominal speed. 

The efficiency was obtained as a function of the flow ratio: actual flow over the maximum flow. Figure 9 shows the efficiency curve of the real-scale prototype installed in Barreda hydroelectric plant.

Leakage losses were also obtained for this real prototype. The real-scale prototype showed similar behavior to the small-scale prototype in experimental tests. Figure 10 shows the percentage of leakage losses in the real-scale prototype. In this turbine, static leakage losses were around 6%. Although the leakage losses for low flows were higher than those in small-scale prototypes, the increase of losses was smaller than in the small-scale prototype. For the small-scale prototype, the leakage losses at the design flow were around 10% (see Figure 8). However, the real-scale prototype had leakage losses below 9% (see Figure 10). 

## 4. Discussion

Numerical studies were used to explain how to design efficient ASTs. These numerical studies led us to understand that ASTs also have buckets and that their filling level is relevant to their design. An analogy between ASTs and bucket elevators was used to understand this concept. 

The efficiency curves of the prototypes studied in this work were compared with previously published curves from previously installed ASTs [11,12,13]. Figure 11 shows a comparison of efficiency curves for five ASTs already installed. The ASTs designed in this work provided high efficiencies even for low flow percentages. For a flow ratio around 20%, the efficiency improved by between 5% and 10% in comparison with other ASTs. This improvement in hydraulic efficiency provides a significant increase in the total efficiency of these turbines.

The percentages of leakage losses for several ASTs were also studied. Figure 12 shows a comparison of the percentages of leakage losses as a function of the percentages of flow running through the turbines. The graph shows very similar static losses for all the ASTs, ranging from 5% to 8%. However, ASTs usually operate under higher flow percentages, above 50%, because their efficiency is higher. For a percentage of flow above 50%, the leakage losses of other designs [11,12,13] increased up to 16%. The ASTs designed in this work caused maximum losses of 10% at the maximum flow. This was due to the design based on the filling level of the AST buckets, which must be around 85% full to reduce overflow leakages. This consideration, as well as the speed control, significantly reduced the leakage losses and consequently improved the efficiency of the turbine.

Figure 12 shows a comparison of leakage loss percentages between previous ASTs and the ASTs designed and installed in this study. The trendlines are in good agreement with the actual data calculated. Figure 12 reveals that static leakage losses were almost the same in all the ASTs studied, around 5% of leakage losses. This value depends on the gap between the flights and the trough of the AST. 

In this study, it was also proved that the reduction of leakage losses in the design of an AST is an important factor for efficiency. The higher the losses due to leakage, the lower the efficiency is. 

To minimize the slope of the leakage losses curve, the AST must be designed to work with the buckets at a filling level of around 85%. In this work, this was obtained using an active speed control which varied the speed as a function of the flow ratio. As shown in Figure 12, the lower slopes of the leakage losses curves were given by the prototypes developed in this work, which present the highest efficiency for any flow ratio in Figure 11. 

A relevant contribution of this study is the significant improvement in efficiency for low flow ratios ranging between 15% and 40%. 

## 5. Conclusions

The main conclusions of this work are listed as follows:
Although buckets of the AST were not identified easily, the analogy with the bucket elevator presented in this work helps in understanding this concept. This analogy was a relevant contribution because the capacity of the buckets of an AST is one of the keys for an efficient design.Although the AST is a volumetric mechanism which works under atmospheric pressure, its efficiency strongly depends on the filling of the buckets. 100% filling would provide the highest efficiency in ideal ASTs. However, because of the movement of the screw, overflow losses occur above 85% filling and the efficiency of the turbine decreases.To ensure 85% filling of the buckets, an active speed control was suggested to ensure that the rotation speed decreased with the decrease of flow.Leakage losses, which are one of the most influential parameters on the efficiency of an AST, can be defined through the static leakage losses, the rotation speed and the design flow. This hypothesis was checked for other ASTs and for the two prototypes studied in this work.

Finally, this work demonstrates how an efficient design of this kind of facility is needed to ensure sustainable generation of energy. Green energy is needed to meet the goals of the 2030 Climate Target Plan but efficient methodologies for the generation of green energy are needed to ensure the sustainability of facilities and infrastructures.

## Figures and Tables

**Figure 1 ijerph-17-09236-f001:**
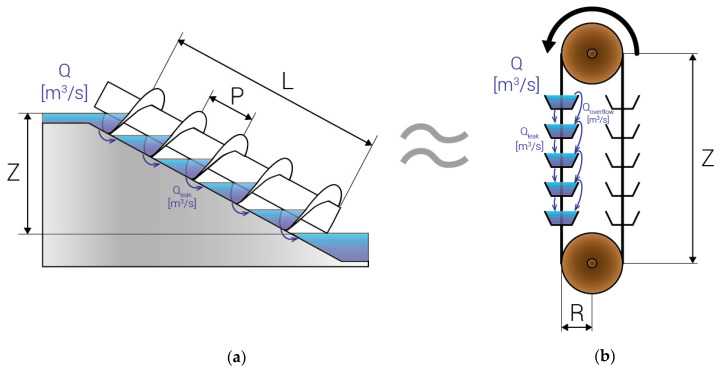
Analogy between Archimedean screw turbine (AST) and bucket elevator. (**a**) AST; (**b**) bucket elevator.

**Figure 2 ijerph-17-09236-f002:**
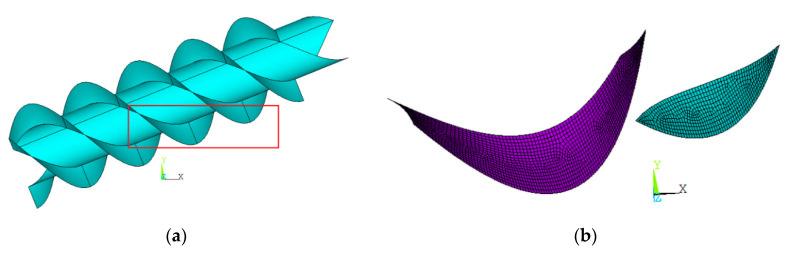
Finite element method (FEM) model. (**a**) Geometry of the AST; (**b**) numerical model of the bucket.

**Figure 3 ijerph-17-09236-f003:**
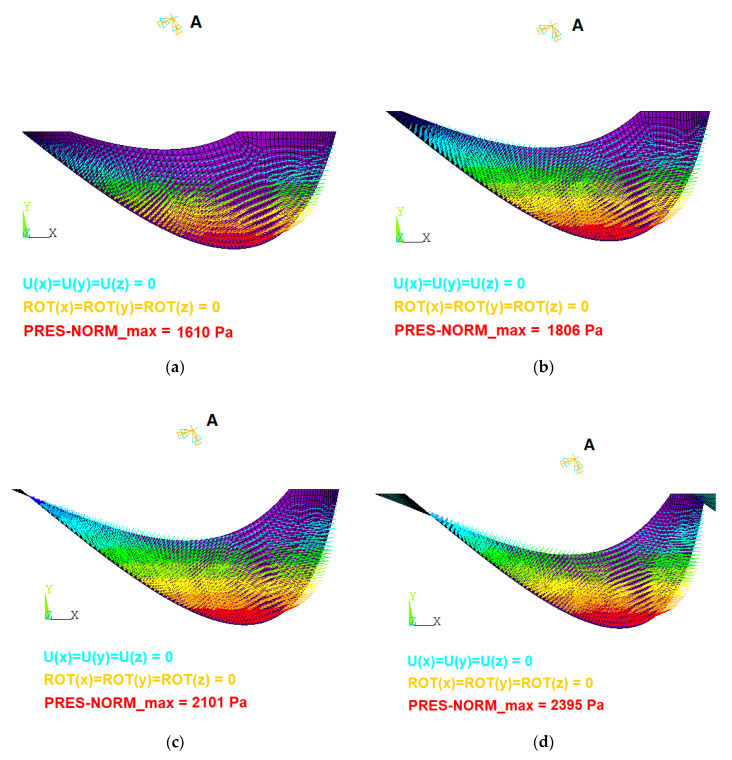
Boundary conditions of FEM model. (**a**) 49% full; (**b**) 52% full; (**c**) 69% full; (**d**) 85% full.

**Figure 4 ijerph-17-09236-f004:**
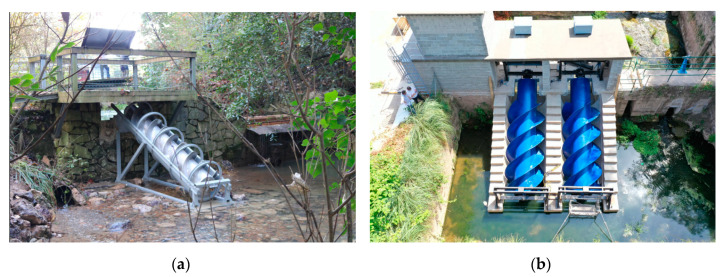
AST experimental prototypes. (**a**) Atlantic Botanic Garden of Gijón (Spain); (**b**) Barreda hydroelectric plant in Torrelavega (Spain).

**Figure 5 ijerph-17-09236-f005:**
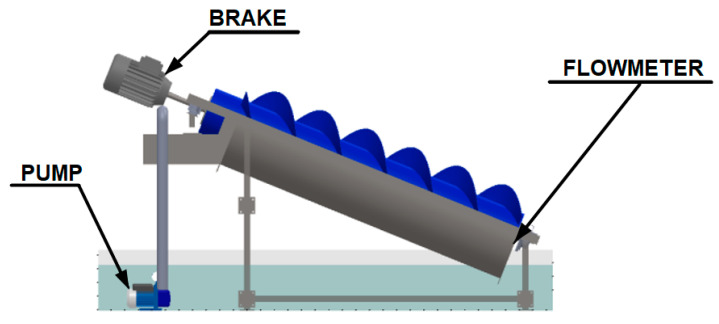
Experimental setup design of the recirculating system.

**Figure 6 ijerph-17-09236-f006:**
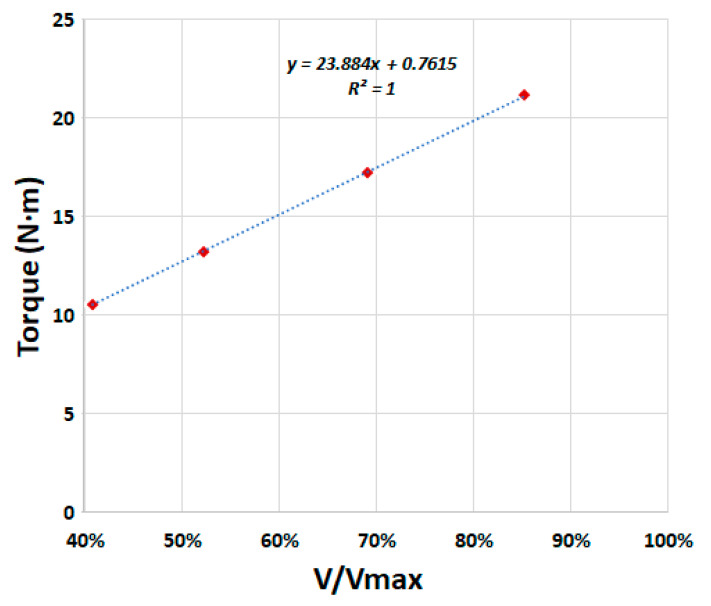
Torque obtained for four filling levels.

**Figure 7 ijerph-17-09236-f007:**
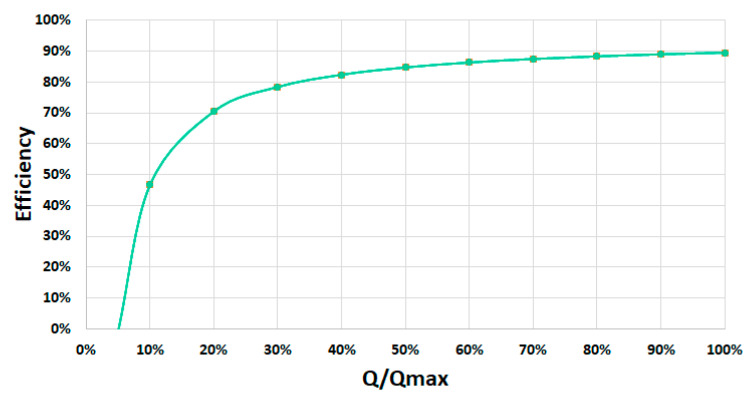
Efficiency curve of the small-scale AST prototype in the Atlantic Botanic Garden (Gijón, Spain).

**Figure 8 ijerph-17-09236-f008:**
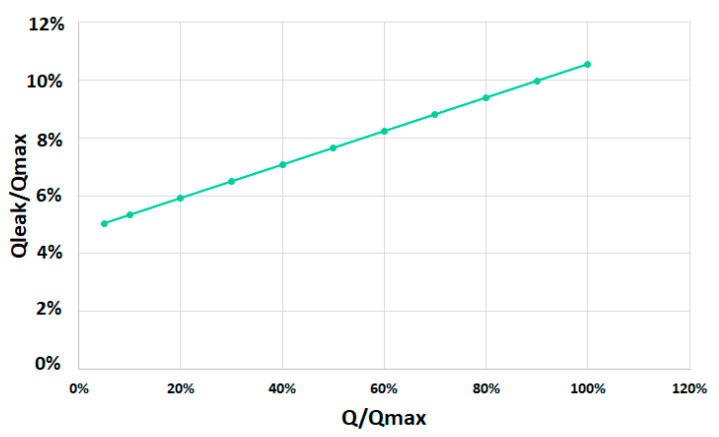
Percentage of leakage losses over the percentage of flow through the screw in the small-scale prototype.

**Figure 9 ijerph-17-09236-f009:**
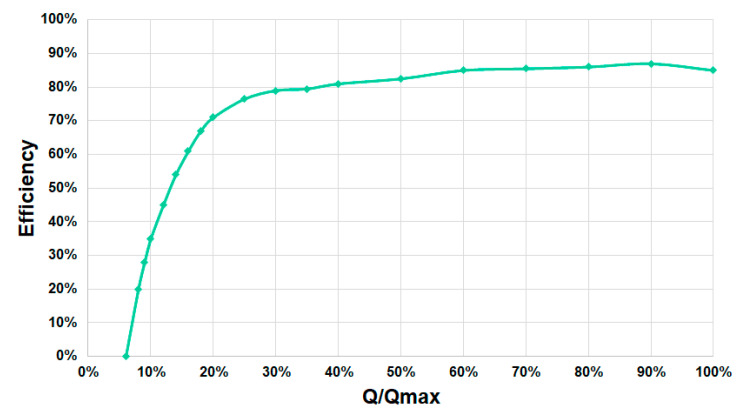
Efficiency curve of the real-scale AST prototype in Barreda hydroelectric plant (Torrelavega, Spain).

**Figure 10 ijerph-17-09236-f010:**
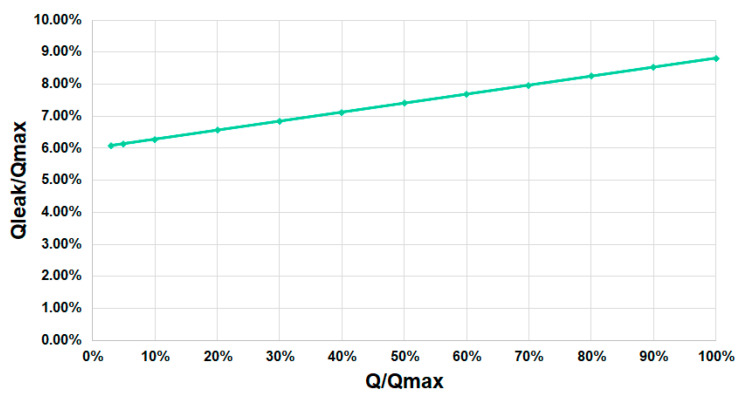
Percentage of leakage losses over the percentage of flow through the screw in the real-scale prototype.

**Figure 11 ijerph-17-09236-f011:**
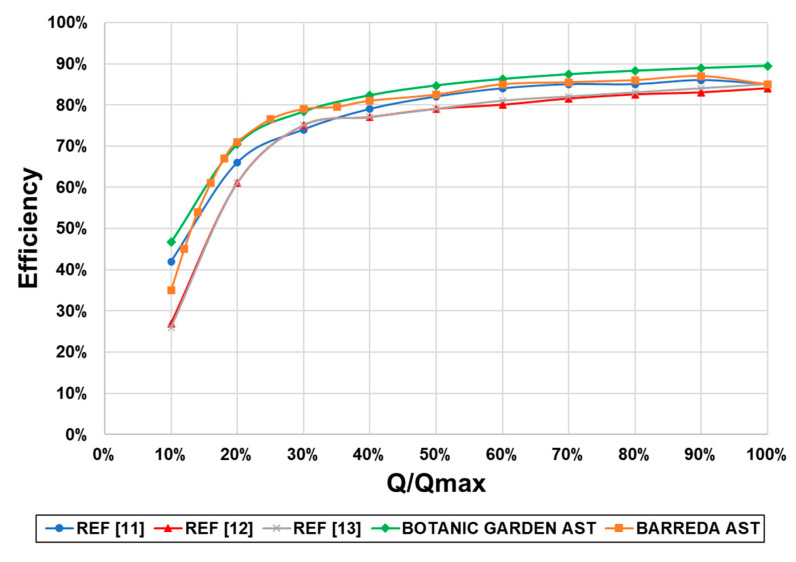
Comparison of previously published AST efficiency curves [11,12,13] and the two prototypes of this work.

**Figure 12 ijerph-17-09236-f012:**
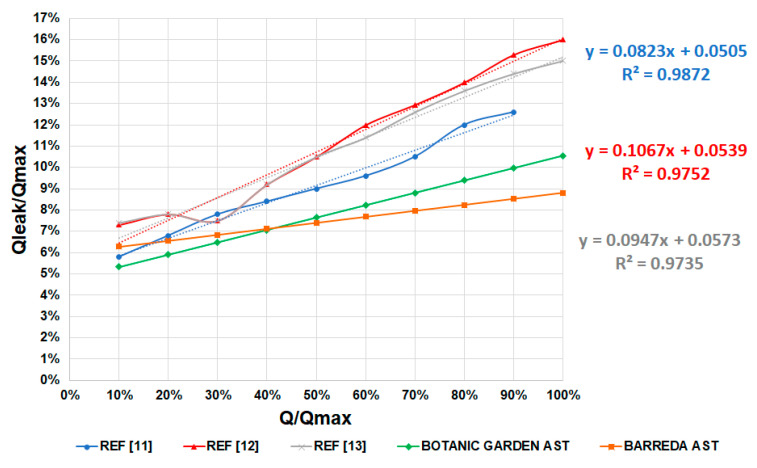
Comparison of the percentages of leakage losses for ASTs from three references [11,12,13] and the two experimental prototypes from this work.

**Table 1 ijerph-17-09236-t001:** Values of the main parameters of the AST numerically studied.

Parameter (Units)	
Slope (°)	22
Outer diameter (m)	0.564
Inner diameter (m)	0.302
Pitch (m)	0.972
Length of the screw (m)	3.203
Head (m)	1.20
Gap between flights and trough (mm)	3

**Table 2 ijerph-17-09236-t002:** Values of the main parameters of the small-scale AST installed in the Atlantic Botanic Garden.

Parameter (Units)	
Slope (°)	22
Outer diameter (m)	0.564
Inner diameter (m)	0.302
Pitch (m)	0.972
Length of the screw (m)	3.203
Head (m)	1.20
Gap between flights and trough (mm)	3
Rotation speed (Ω)	73.239

**Table 3 ijerph-17-09236-t003:** Values of the main parameters of the real-scale AST installed in the Barreda hydroelectric plant.

Parameter (Units)	
Slope (°)	22
Outer diameter (m)	2.278
Inner diameter (m)	1.273
Pitch (m)	4.540
Length of the screw (m)	4.179
Head (m)	1.94
Gap between flights and trough (mm)	3
Rotation speed (Ω)	variable

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
