# Peer review of "A New Methodology to Design Sustainable Archimedean Screw Turbines as Green Energy Generators"

_ijerph, 2020, doi:10.3390/ijerph17249236_

Round 1
Reviewer 1 Report
General comments:
The manuscript evaluated the performance of AST using an analogy between ASTs and bucket elevators, and a new methodology to design ASTs based on leakage losses curve is provided. This study is global relevance with respect to contribute the development of more efficient and sustainable facilities. However, it is not presented in an appropriate and substantiated way. Major revisions are required according to the specific comments given below, and it could be reconsidered after revisions required. My major concern is the superficiality of the writing and the lack of objectivity. Thus, a wide and deep improvement and organization of the manuscript is necessary.
Specific comments:
Introduction
Line 46 – remove: …The AST is efficient in…. It is repeated in line 44
Line 49 – insert reference after friendly…
Line 65-73 – The last paragraph must be more objective. It is needed to be rewritten. The authors must be clearer regarding the purpose of the study. I recommend improving the hypothesis (questions are a good option).
Material and Methods
This section needs a broad revision and improvement. It looks like writing a report and not a scientific paper.
Line 76-81 – It should be removed to introduction.
Line 99 – rewrite: …. The authors of this study … to: Three operation conditions were applied to study AST.
Line 121-126 - This part mainly refers to the objectives of the study and should be removed from this section and inserted at the end of the introduction. And for this, the necessary text adjustments must be made.
Line 135-143 – Introduction????
Line 143-144- It is aim of study. The authors must write methodologies used to achieve the objectives more appropriately.
Figure 2 shows the efficiency curves from previous published AST. This figure should be showed in the results inserting the data of study to compare the curves.
Figure 3 This figure should be showed in the results inserting the data of study to compare them.
Lines 196-197 – These issues must be clear in the objectives of the study in the end of introduction.
Results
Line 229 – 234 – Remove it. It is not necessary.
Line 282 – has instead have
Discussion – The discussion is not too convincing. In part the discussion repeats the results in other words rather than discussing the findings. Remove the repetitive elements of results and discuss critically the new methodology proposed in relation to previous studies and their main findings.
Figure 12 and 13 – they must be moved to results and properly described.
Author Response
Thank you for all the comments. Please see the attachment.

Reviewer 2 Report
Dear Authors,
It was really interesting to me to review your manuscript containing such a thorough technical and mathematical study. The presented research can make a significant contribution to the research area. However, there are minor flaws that prevent me from recommending publication in the present form.
1. The article contains some typos and inaccuracies. Please explain what water21 is in line 139. There seems to be a missing part of the paragraph ending in line 293.
2. The quality of the figures needs to be improved. In particular, in Fig. 7, other X-axis limits can be set.
3. How could you explain the intersections of the curves in fig. 12 and 13? What does this trend indicate?
Thus, my decision is a minor revision.
Author Response

(The authors gave the same response as above.)

Reviewer 3 Report
Nice article, especially that this has been done theoretical and compared with prototypes.
However I have the following comments and suggestions for the improvement.
- Overall the figures with the curve fittings are difficult to read
- Describe more the relation to the global energy problem. How much can these type of turbines help on the overall energy demand and how large is the market potential of these systems, are these types of turbines not a more marked specific solution to specific markets, rather than helping in the overall energy problem.
- It would be could to include and describe the proposed speed control
- Please include also a description of the measurement setup and the experiments.
- It would also be nice to include some general loss estimations of the overall system, including the losses of the electrical generator, converter and mechanical system (bearing losses etc.), etc. and relate the overall losses to the increased efficiency of the design. As the improved efficient is only on the AST. How much does a 10 % efficiency increase the overall system efficiency ?
Author Response

(The authors gave the same response as above.)

Round 2
Reviewer 1 Report
General comments:
The authors met most of my suggestions and recommendations. However, I think that the discussion should be improved. Some revisions are required according to the specific comments given below.
Specific comments:
Introduction
Line 49 - It must be added why AST is considered fish friendly. See in Piper et al. (2018) (https://www.sciencedirect.com/science/article/abs/pii/S0925857418301137) that mentions: ... ASTs are generally perceived as 'fish friendly' due to their slow rotational speed, low shear force and small pressure changes, when compared to conventional Francis and Kaplan turbines (Kibel et al., 2009; Spah, 2001) ...
Discussion - Considering my previous comments, very little has been improved in the discussion. Thus, discuss critically the new methodology proposed in relation to previous studies and their main findings.
Line 326 – To correct the error message after Figure 12
Author Response
General comments:
The authors met most of my suggestions and recommendations. However, I think that the discussion should be improved. Some revisions are required according to the specific comments given below.
Thank you very much for your comments. We have revised the discussion. We have followed the specific comments to improve the quality of the manuscript.
Specific comments:
Introduction
Line 49 - It must be added why AST is considered fish friendly. See in Piper et al. (2018) (https://www.sciencedirect.com/science/article/abs/pii/S0925857418301137) that mentions: ... ASTs are generally perceived as 'fish friendly' due to their slow rotational speed, low shear force and small pressure changes, when compared to conventional Francis and Kaplan turbines (Kibel et al., 2009; Spah, 2001)
Thank you very much. We have included a specific sentence to detail why ASTs are fish-friendly technology in comparison with other generators.
“Due to the low rotational speed and the atmospheric pressure, most fish species are able to pass through the turbine with no injuries.”
Discussion - Considering my previous comments, very little has been improved in the discussion. Thus, discuss critically the new methodology proposed in relation to previous studies and their main findings.
Thank you very much for this comment.
The main findings of this work are that the AST must be designed taking into account two relevant concepts:
- Buckets filling must be around 85%
- Leakage losses must be minimized for any flow ratio.
And these losses are linearly dependent on the flow ratio. Results from this work are compared with previously designed ASTs. Two figures (figure 11 and figure 12) compare results of all of them. Discussion section includes a comparison of results such as:
“ASTs designed in this work provide high efficiencies even for low flow percentages. For a flow ratio around 20% the efficiency improves between 5 and 10 %.”
However, in order to improve the discussion section the suggestion of this reviewer was considered and some changes in the manuscript were done (lines 305 to 308; line 320; and lines 323 to 334)
Line 326 – To correct the error message after Figure 12
Thank you. The error message has been removed.
Reviewer 3 Report
Thanks, for the improved article and the response to my comments.
It clarifes my questions and also the updated article has been improved and can be submitted.
There are some cross references which are missing.
Author Response
Comments and Suggestions for Authors
Thanks, for the improved article and the response to my comments.
It clarifes my questions and also the updated article has been improved and can be submitted.
There are some cross references which are missing.
Thank you very much for your comments. We have revised all the references. Error in line 326 has been removed.